# Automatic Text-Mining Approach to Identify Molecular Target Candidates Associated with Metabolic Processes for Myotonic Dystrophy Type 1

**DOI:** 10.3390/ijerph20032283

**Published:** 2023-01-27

**Authors:** Dhvani H. Kuntawala, Filipa Martins, Rui Vitorino, Sandra Rebelo

**Affiliations:** Medical Sciences Department, Institute of Biomedicine—iBiMED, University of Aveiro, 3810-183 Aveiro, Portugal

**Keywords:** myotonic dystrophy type 1, metabolism, bibliometric analysis, VOSviewer, bioinformatics, functional enrichment analysis

## Abstract

Myotonic dystrophy type 1 (DM1) is an autosomal dominant hereditary disease caused by abnormal expansion of unstable CTG repeats in the 3′ untranslated region of the myotonic dystrophy protein kinase (*DMPK*) gene. This disease mainly affects skeletal muscle, resulting in myotonia, progressive distal muscle weakness, and atrophy, but also affects other tissues and systems, such as the heart and central nervous system. Despite some studies reporting therapeutic strategies for DM1, many issues remain unsolved, such as the contribution of metabolic and mitochondrial dysfunctions to DM1 pathogenesis. Therefore, it is crucial to identify molecular target candidates associated with metabolic processes for DM1. In this study, resorting to a bibliometric analysis, articles combining DM1, and metabolic/metabolism terms were identified and further analyzed using an unbiased strategy of automatic text mining with VOSviewer software. A list of candidate molecular targets for DM1 associated with metabolic/metabolism was generated and compared with genes previously associated with DM1 in the DisGeNET database. Furthermore, g:Profiler was used to perform a functional enrichment analysis using the Gene Ontology (GO) and REAC databases. Enriched signaling pathways were identified using integrated bioinformatics enrichment analyses. The results revealed that only 15 of the genes identified in the bibliometric analysis were previously associated with DM1 in the DisGeNET database. Of note, we identified 71 genes not previously associated with DM1, which are of particular interest and should be further explored. The functional enrichment analysis of these genes revealed that regulation of cellular metabolic and metabolic processes were the most associated biological processes. Additionally, a number of signaling pathways were found to be enriched, e.g., signaling by receptor tyrosine kinases, signaling by NRTK1 (TRKA), TRKA activation by NGF, PI3K-AKT activation, prolonged ERK activation events, and axon guidance. Overall, several valuable target candidates related to metabolic processes for DM1 were identified, such as NGF, NTRK1, *RhoA*, *ROCK1*, *ROCK2, DAG, ACTA, ID1, ID2 MYOD***,** and *MYOG*. Therefore, our study strengthens the hypothesis that metabolic dysfunctions contribute to DM1 pathogenesis, and the exploitation of metabolic dysfunction targets is crucial for the development of future therapeutic interventions for DM1.

## 1. Introduction

Myotonic dystrophy type 1 (DM1), also known as Steinert’s disease, is the most common type of muscular dystrophy in adults, with a prevalence of approximately 5 to 20 per 100,000 individuals worldwide [1,2]. DM1 is a multisystemic neuromuscular disorder mainly characterized by myotonia, muscle weakness, and atrophy. Other features observed in patients with DM1 include alterations in the central nervous system (CNS), gonadal atrophy, insulin resistance, dyslipidemia, cardiac conduction deficiencies, and breathing difficulties [3]. This disorder is caused by a trinucleotide expansion of unstable repetitions of CTG in the 32,032 untranslated region of the myotonic dystrophy protein kinase (*DMPK)* gene located at chromosome 19 q13.3 [4,5]. Fewer than 38 CTG repeats are found in normal individuals, whereas DM1 patients have at least 50 CTG repeats [6]. Interestingly, the age of onset seems to be inversely correlated with CTG repeat length, influencing the severity of clinical symptoms [7]. Despite research efforts in the field, there is currently no cure for DM1, and diagnosis is also challenging to halt or slow down DM1 progression.

Metabolic disorders are known to result from metabolism problems, which affect neurodegeneration, longevity, and aging [8,9]. Various metabolic defects, including glucose resistance, hyperinsulinemia, and diabetes mellitus, are found in DM1 patients [10]. Moreover, whereas DM1 diagnosis using genetic tests and magnetic resonance, electromyography, and skeletal muscle histopathology are standard methodologies [11], creatine kinase is also known to be the only reasonable biochemical marker in DM1 patients, although not a disease-specific marker [12,13]. Few studies have been conducted regarding the advance of non-invasive biomarkers for DM1, but promising results have been reported on the use of microRNAs (miRNAs) [14]. miR-1, miR-27b, miR-133a, miR-133b, miR-140-3p, miR-206, miR-454, and miR-574 have been reported to be deregulated in DM1 patients compared to healthy individuals [15,16,17]. Additionally, promising treatments for DM1 have been proposed, which might be able to suppress or eliminate its associated molecular effects, including *MBNL1* (Muscleblind Like Splicing Regulator 1) expression enhancers, and toxic RNA degradation [18]. It was also recently reported that treatment of DM1 patient-derived fibroblasts with metformin reversed metabolic and mitochondrial defects, including impaired proliferation [19]. Metformin has the potential to delay aging at both the cellular and organismal level, and metformin treatment may show efficiency in a preclinical setting [20]. Therefore, an in-depth study of the molecular mechanisms and therapeutic targets in DM1 has become a key research topic. However, the abovementioned approaches are not currently treatment options for DM1. Therefore, approaches to understand the contribution of metabolism and mitochondrial dysfunction in DM1 seems to be crucial to unraveling DM1 pathogenesis. However, only few studies have been carried out to identify novel putative metabolic and therapeutic targets for DM1. In the present work, we aimed to identify novel metabolism-associated molecular targets, resorting to bibliometric and visualization-based analyses, together with bioinformatics, to map the biological processes and associated signaling pathways. These potential novel molecular targets will hopefully highlight a new perspective to address research gaps in DM1 pathogenesis.

## 2. Material and Methods

### 2.1. Search Approach

Literature searches were performed in the Web of Science (WOS) and Scopus databases. The Scopus database was chosen over the WOS because more results were retrieved, and information is updated on a daily basis [21,22]. The first search query included the following terms: “myotonic and dystrophy and type 1” and “metabolic or syndrome”. The second query included the following terms: “myotonic and dystrophy and type 1” and “metabolic or metabolism or skeletal or muscle or metabolism”. Both queries aimed to identify scientific articles related to metabolism and DM1. These search queries included studies from the 1960s onwards, and the generic terms “review”, “animals”, and “non-English” were excluded from the search tab.

### 2.2. Automatic Text-Mining Analysis Using VOSviewer

VOSviewer is a software tool used to reveal scientific landscapes and to show associated terms from selected scientific articles through a network [23]. To extract the keywords related to DM1, we used VOSviewer (v. 1.6.16), whereby the layout was established in a context for mapping and clustering. To create the bibliometric networks, a co-occurrence analysis of all closely related terms was carried out to indicate how closely keywords were associated by computing and extracting keywords, in addition to numerical evaluation of the content of a group of documents. Usually, if keywords are grouped into the same cluster, they likely to reflect similar topics. Thereafter, these results can be analyzed to build a conceptual structure of the desired field [24,25]. We aimed to determine the co-occurrence of keywords between different clusters. All the metadata from the articles were uploaded by combining the keywords included in queries 1 and 2. These include “myotonic and dystrophy and type 1” and “metabolic or syndrome or metabolism or skeletal or muscle”. Generic terms such as ‘follow-up’, ‘article’, ‘review’, ‘male’, ‘female’, and many other irrelevant terms were removed by manual curation in VOSviewer. Thereafter, networks were created based on the relation strength of the search terms and keywords from the different clusters, which were displayed in different colors. Possible molecular associations were depicted if a specific keyword had a simultaneous association with DM1 or any identical keywords by setting a minimum number of occurrences of a keyword to 63. Based on this analysis, we anticipated that we would be able to identify popular research topics in DM1 research.

Furthermore, to retrieve a gene list from VOSviewer, we combined the total 8 keywords from both queries using Scopus search. The resulting files were uploaded, and we manually ticked the genes/proteins present in this list in VOSviewer. We manually curated duplicates and plural names of genes/proteins from the final list to reduce redundancy. The UniProt Knowledgebase (UniProtKB) website [26] was used to review entries and obtain the gene/protein identifiers. Only the reviewed (Swiss-Prot) entries that contained manually annotated records were chosen. Each identifier was recorded manually for further analyses. However, some of the genes downloaded from Scopus had no UniProt identifier because the gene was broad. Therefore, the nearest gene/protein was chosen to explore the possibility of any similarities.

### 2.3. DisGeNET Search

To complement the VOSviewer results (Section 2.2) and identify the genes previously associated with DM1, the DisGeNET database (version 7.0) was searched. DisGeNET is an open-access management program that combines information on genes and gene variants related to human diseases from numerous resources with data obtained by text mining of scientific information [27,28]. We searched for “Myotonic dystrophy type 1” (C3250443 code) in the query box and opted for a summary of gene–disease associations, keeping the default query focused on diseases. The jvenn plug-in tool [29] was used to compare the lists of results obtained from DisGeNET and VOSviewer.

### 2.4. Protein–Protein Interaction Network Construction

Protein–protein interaction (PPI) network analysis was performed using the STRING web tool (version 11.5) [30] to illustrate the physical and functional association between the genes retrieved in the VOSviewer analysis and that were not previously associated with DM1. The interaction score was kept at the highest confidence (0.900) to easily view the strength connection between the genes.

### 2.5. Gene Ontology Functional and Pathway Enrichment Analyses

A functional enrichment analysis was performed using the g:Profiler tool (version e106_eg53_p16_65fcd97), which maps genes to functional data sources and identifies statistically notable gene ontology (GO) terms [31]. GO enrichment analysis (GO biological process) and REAC signaling pathway analysis were performed. The default parameters of the enrichment analysis were as follows: human was chosen as a specific organism, and the user threshold was 0.05.

## 3. Results

Considering the aim of the present manuscript, a literature search was performed, followed by automatic text mining and data functional enrichment analysis to identify relevant associations between metabolism/metabolic processes and DM1. A summary of the research strategy used in the present manuscript is represented in Figure 1.

### 3.1. Literature Search and Automatic Text-Mining Analysis to Unveil Novel Metabolism Molecular Targets in DM1

The Scopus literature search yielded 12 497 papers using both queries after excluding articles on animals, as well as non-English and review articles. After manual curation by the author to remove duplicates from both queries, a total of 7599 papers were obtained (Figure 1). VOSviewer software was used to perform a text-mining analysis of the article data by construction and visualization of the co-occurrence networks of relevant terms (Figure 1).

Using 45 co-occurrences as a threshold in VOSviewer software settings, 863 terms were obtained and grouped into 4 clusters represented by different colors (Figure 2). The size of nodes reflects the frequency of appearance of the keywords, and the distance between two nodes represents their correlation. Cluster 1 (red) is related to metabolism and includes keywords such as gene expression, protein binding, and RNA-binding protein (Figure 2A), with the second highest total link strength (Table 1) after myotonic dystrophy. Cluster 2 (green) is related to myotonic dystrophy and includes keywords such as myotonia, muscular dystrophy, myotonic dystrophy type 1, and myopathy (Figure 2A). Cluster 3 (blue) is related to trinucleotide repeat, nucleotide repeat, gene mutation, genetic analysis, and pathogenesis (Figure 2A). Cluster 4 (yellow) includes keywords such as skeletal muscle, muscle fibers, skeletal, muscle cell, and calcium (Figure 2A). Subnetworks around the terms “Metabolism” (Figure 2B) and “Myotonic Dystrophy” (Figure 2C) were created to analyze the associated terms. Overall, it is clear that the keywords metabolism and myotonic dystrophy are represented as large nodes with a high total link strength, suggesting the significance of these keywords, followed by gene expression, skeletal muscle, muscle skeletal, and RNA-binding proteins.

Table 1 summarizes the top six keywords according to their total link strength relative to the global network of all keyword co-occurrences (Figure 2A). Table 1 also indicates that the keyword myotonic dystrophy has the highest link strength, and RNA-binding proteins has the lowest strength.

In order to identify the genes related to DM1 in VOSviewer, using the default settings, 39,565 terms were found to meet the threshold by setting the minimum number of occurrences to 1. By default, 5 clusters with a total of 113 genes (Appendix A). Among the 113 genes, ‘amino acids, drosophila, duplicates, plurals and generic terms’ were manually curated from the list, and a total of 60 genes were attained. Furthermore, to obtain relevant information about these 60 genes, closely associated genes were further retrieved from the UniProt database, and 26 genes were added (Appendix A). Thereafter, a total of 86 genes was considered for further analysis (Figure 1).

### 3.2. Comparative Analysis of the Novel Identified Metabolism-Associated Molecular Targets in DM1 Using VOSviewer with Molecular Associations Previously Described in DisGeNET

In order to compare the novel identified metabolism targets associated with DM1 (Appendix A) with the genes/proteins already associated with DM1 in the literature, a search of the DisGeNET database was performed. A total of 179 genes associated with DM1 were retrieved (Appendix A). A comparative analysis using the jvenn tool revealed that 15 common genes: *DMPK, MBNL, CELF1, INSR, IL6, IGF1, DMD, PRKCA, ACTB, ATP2A1, ATP2A2, RYR1, CASP3, TGFB1*, and *CKM* (Figure 3 and Table 2).

For further analysis, we excluded these 15 common genes between VOSviewer and DisGeNET, and a list of 71 candidate genes associated with metabolism was obtained.

### 3.3. Characterization and Functional Enrichment Analysis of the Novel Identified Metabolism Associated Molecular Targets in DM1

Given that the main goal of the present manuscript is the identification of novel putative metabolism-associated targets for DM1, we next focused our analysis on the 71 genes exclusively identified using the VOSviewer analysis (Figure 1). First, we gathered information regarding its subcellular localization, molecular function, and biological process using ProteomicsDB and UniProt (Table 3 and Appendix A). More importantly, their dysfunction in DM1, if already reported, or gene information retrieved through the NCBI (National Center for Biotechnology Information) (Table 3) was searched, as these genes may reveal a great potential as metabolism-associated targets for DM1. 

Furthermore, we analyzed the protein–protein interactions (PPI) of these candidate targets using the STRING database. The resulting network outlines the arrangement of anticipated associations for the 71 genes considering all possible protein interactions, which comprised 69 nodes (genes) interacting with each other through 60 edges, showing a significant PPI enrichment p-value: 1.53 × 10^−14^; therefore, these genes are possibly biologically connected (Figure 4). The most interactions were observed for *INS* and *NTRK1* proteins. *AKT1, CDC42,* and *LAMA1* proteins also presented a higher number of interactions.

Moreover, using g:Profiler, we performed a functional enrichment analysis in terms of biological processes (BPs) and pathways of the 71 candidate molecular targets for DM1 (Figure 5). A total of 187 GO terms were found to be enriched in the BP category (Appendix A). The candidate molecular targets for DM1 were mainly associated with the regulation of primary metabolic processes, regulation of cellular metabolic processes, regulation of nitrogen compound metabolic processes, gene expression, and cell differentiation (Figure 5A). Furthermore, several signaling pathways were found to be associated with the candidate molecular targets for DM1 using the REAC database (Appendix A), from which signaling by receptor tyrosine kinases, axon guidance, and *NTRK1* (TRKA) presented a higher number of associated genes/proteins (Figure 5B).

Based on the previous analysis, it was evident that biological processes associated with metabolic processes are overrepresented. Therefore, we compiled a list of the genes associated with metabolic processes, which comprises 44 genes/proteins. The PPI network comprising the 44 identified proteins is presented in Appendix A. Five clusters were constructed using the STRING database by applying the Markov clustering (MCL) algorithm after summarizing the complied list of genes associated with metabolic processes (Figure 6). Cluster 1 includes the *RHOH, AKT1*, *ACTA1, ACTN1, HSPB1, HSPA1A*, and *MAPT* genes. Two chaperones (*HSPB1* and *HSPA1A*) were present in this cluster. Cluster 2 includes the *MYOD1, ID1, ID2, TCF3*, and *MYOG* genes. Cluster 3 comprises the *NGF, NTRK1, PIK3CA*, and *INS* genes. It was possible to verify that cluster 3 comprises genes known to be important signaling pathways, such as PI3K/AKT activation and signaling by NTRK1. Lastly, cluster 4 includes the *FXR1, FXR2*, and *MAPK1* genes, whereas cluster 5 includes the *ROCK1* and *ROCK2* genes. As anticipated, this cluster-based functional characterization of the metabolic gene list unveiled specific physiological roles/signaling pathways for several subgroups of interacting genes (metabolic processes and myogenesis/muscle contraction) (Figure 6).

## 4. Discussion

DM1 is the most prevalent form of muscular dystrophy among adults, causing multisystemic symptoms in patients and with no treatment currently available. In the present work, we offer a comprehensive analysis based on a bibliometric approach to identify novel candidate targets for DM1 related to metabolism. Upon a systematic literature search, the work was divided into four steps: analysis of keyword co-occurrences, selection of candidate gene targets, analysis of the shortlisted targets, and analysis of biological and signaling pathways associated with the obtained gene list. Interestingly, the term metabolism frequently co-occurs with terms such as gene expression, protein binding, protein phosphorylation, and protein function (Figure 2B). This is not unexpected, given that in previous work, our described that metabolic and mitochondria dysfunction contribute to DM1 pathophysiology [57]. In fact, there are several pieces of evidence of metabolic defects in DM1 patients, such as insulin resistance, hyperinsulinemia, and diabetes mellitus [34]. Furthermore, previous work by our team associated these metabolic alterations with lipins; phosphatidate phosphatase enzymes, which regulate the lipid signaling pathways; DAG levels; triacylglycerol levels; and phospholipids [57].

Myotonic dystrophy also co-occurs with keywords that are associated with main clinical features observed in patients with DM1, such as muscle weakness, myopathy, myotonia, and muscle contraction (Figure 2C), which is in line with other previous studies [102,103].

Further analysis of automatic text mining carried out in VOSviewer in combination with DisGeNET data showed an intersection of 15 genes that were common between the two software programs, of which *DMPK, MBNL, CELF1* are important players in DM1 pathology and are the most studied in DM1 [104,105,106]. Other common genes include *INSR, IL6, IGF1, DMD, PRKCA, ACTB, ATP2A1, ATP2A2, RYR1, CASP3, TGFB1,* and *CKM* (Table 2). Remarkably, with this analysis, we identified 71 genes not previously associated with DM1. Among these, several valuable candidate targets for DM1 related to metabolic process emerged (Figure 6, Table 3).

The last part of our work focused on performing a biological process and pathway enrichment analysis of the 71 genes not previously associated with DM1. It was evident that more genes participate in regulation of cellular metabolic process, regulation of primary metabolic process, and regulation of nitrogen compound metabolic processes (Figure 5). Furthermore, cluster-based functional characterization of the metabolic gene list grouped the genes in five different and highly relevant clusters (Figure 6), which are discussed below. Additionally, a novel automated string-based approach for cluster analysis is a possible alternative and an interesting approach to that used in the present study [107].

### 4.1. Dysregulation of Biological Processes and Signaling Pathways Associated with DM1

#### 4.1.1. Muscle Function and Associated Pathways

Cluster 1 comprises genes important for skeletal muscle (*ACTA1*) and muscle function (*ACTN1*) and includes a crucial gene (AKT1) known to regulate insulin signaling, cell survival, and tumor progression (Table 3), as well as two chaperones: *HSPA1A* and *HSPB1* (heat shock proteins (HSPs)). HSPs are known to be expressed in skeletal muscle and play key roles in muscle growth and development [108]. Both *HSPA1A* and *HSPA1L* are known as HSP70 or commonly as HSP72 [109]. HSP70 is stimulated in response to non-damaging and damaging stress stimuli. Hence, its overexpression leads to the maintenance of muscle fiber cohesion and enables muscle reconstruction and repair. However, HSP70 expression is reduced during muscle aging, and the loss of HSP70 as a vital mechanism may lead to contractile dysfunction and muscle atrophy [110]. HSP70 upregulation in skeletal muscle has been recognized in rodent models of muscle injury, muscle atrophy, and muscular dystrophy, emphasizing the role of HSP70 as an important therapeutic target for the treatment of different disorders that have a negative impact on skeletal muscle function [111]. Likewise, glycogen synthase kinase 3 beta (GSK3β) is a familiar conserved serine/threonine kinase with roles in pathways regulating myogenesis, inflammation, neurogenesis, metabolism, and cellular processes in skeletal muscle [112]. Although evidence shows a vital role for GSK3β in skeletal muscle metabolism, it is also involved in the development of clinical conditions. Notably, increased activity of GSK3 is described in DM1 [113]. In fact, it was shown that GSK3β inhibition improved defects in myotonia, myogenesis, and muscle strength in a DM1 mouse model [114]. It was also shown that normalization of CUGBP1 activity with GSK3 inhibitors had a positive effect on reducing skeletal muscle and CNS morphology in DM1 mouse models [115]. Overall, GSK3β can be pointed to as a therapeutic target [114] for the treatment skeletal muscle wasting induced by aging and a number of chronic diseases [116].

#### 4.1.2. NGF and TRK Signaling Pathways

Moreover, in the PPI analysis network (Figure 4), a strong degree of interaction was revealed between *INS* and *NTRK1*, which was further evidenced by cluster analysis (cluster 3, Figure 6). NTRK genes comprise *NTRK1*, *NTRK2*, and *NTRK3*, encoding the proteins of tropomyosin receptor kinase (TRK) family TRKA, TRKB, and TRKC, respectively, which are transmembrane tyrosine kinases receptors. The latter receptors have been associated with several functions, including precursor cell survival and proliferation, differentiation, metabolism, sensory neuron function, synaptic strength, and plasticity [117]. These tyrosine kinase receptors have received some attention from the clinical point of view, given that *NTRK* gene fusions including *NTRK1*, *NTRK2,* and *NTRK3* are identified as oncogenic drivers in various types of tumors, including an increase in cancer in DM1 [118]. The *NTRK1* gene encodes the TRKA protein, which binds to the nerve growth factor (*NGF*), inducing tyrosine phosphorylation and tyrosine kinase activity of TRKA [119]. Furthermore, the ligand binding to TRK receptors causes TRK receptor dimerization and activates three main intracellular signaling pathways, namely phospholipase C-γ (PLCγ), PI3 kinase (*PI3K*), and mitogen-activated protein kinase (MAPK/ERK) pathways (Figure 7) [120]. The ROCK pathway can also be activated both via the Rac-RhoA or the Raf-MAPK-RSK signaling pathway [121,122,123,124]. Together, these pathways play important and distinct roles in cell functioning. The MAPK/ERK pathway is involved in cell growth and proliferation, whereas the PLCγ pathway regulates neuronal differentiation, survival, and metabolism. The PI3K pathway is responsible for metabolism, survival, and apoptosis prevention [125]. Interestingly, crosstalk between these signaling pathways occurs to coregulate biological functions mediated by *NTRK* genes. The appropriate activation of TRK receptors is critical to nervous system development and cell survival.

In the present work, eight genes associated with the TRK signaling pathways were identified as being related to DM1 (Figure 7, indicated in blue). Among these, only four were previously reported as altered in DM1, namely *ERK1/2*, *PI3K, AKT1*, and *PKC* (Figure 7). The MEK/ERK pathway mainly promotes and regulates cellular proliferation [126], and in DM1, it is abnormally active in the early stages of myoblast differentiation [127]. Additionally, the aberrant expression of other cell proliferation stimulators observed in DM1, such as protein kinase R (PKR), protein kinase R-like ER kinase (PERK), and pyruvate kinase M2 (PKM2), is also observed in cancer, which may explain why DM1 patients present with increased cancer susceptibility [128,129,130].

AKT and mTOR, which are important stimulators of anabolic pathways such as glucose uptake, glycogen storage, and protein synthesis did not respond to insulin stimulation and were therefore found to be decreased in DM1. In addition, the MEK/ERK pathway is responsible for the growth-promoting effects of insulin [131]. Together, they regulate several biological processes, such as transcription, protein synthesis, cell growth, and differentiation [132]. *ERK1/2* was also found to be decreased in DM1 upon insulin stimulation [133].

We hypothesized that NGF signaling and downstream pathways are highly relevant to the pathophysiological mechanism of DM1, i.e., *NGF, NTRK1*, *RhoA*, *ROCK1/2*, and DAG novel putative DM1 targets. The latter is a second messenger, and its levels are correlated with PKC and PKD levels, given that these kinases are DAG sensors. Aberrant activation of PKC and PKD contributes to the development of metabolic diseases. Given that PKC is upregulated in DM1, we hypothesized that DAG is also increased in DM1, accumulating in many organs and leading to metabolic homeostasis disruption. The follow-up of DAG levels in DM1 is of paramount importance, given that it could be used as DM1 biomarker and as a disease progression biomarker.

The role of downstream TRK receptor signaling pathways has been intensively studied with respect to neuronal survival. The truncated TRK receptors, which are most abundant in non-neuronal tissues, seem to be important for skeletal muscle, as their presence is well documented during late developmental, early postnatal, and adult stages. However, TRK levels during myoblasts division, differentiation, and myotube formation seems very low. Conversely, p75NTR seems to be the most abundant neurotrophic receptor during early development, and receptor complexes involving both p75NTR and TRK might occur only in the late stages of development [134].

Furthermore, the impact of neurotrophins on muscle cell differentiation has been investigated both in vitro and in vivo. Essentially, an effect on differentiation and a trophic antiapoptotic effect of NGF in early myotubes have been suggested [134]. It was demonstrated that NGF signaling, through its low-affinity p75NTR receptor, is mediated by *RhoA* in muscle cells and is required for physiological myoblast fusion and to maintain a functional cytoskeletal organization of myotubes [135]. These results are particularly relevant in vivo when fibers are damaged, given that the activation of myogenic precursors is needed, as well as their fusion with existing myofibers, contributing for the formation of fibers with a functional contractile unit, allowing for efficient repair of damaged areas of skeletal muscle [135].

#### 4.1.3. Insulin Signaling Pathways

As mentioned above, there is crosstalk between insulin-mediated signaling and NGF signaling. In our PPI analysis, we observed interactions of insulin with *NGF, NTRK1, PI3K*, and *AKT1* (Figure 4). This is particularly interesting, given that insulin exerts its biological functions by binding to insulin binding receptors such as insulin receptor (IR) and insulin growth factor (IGF). Besides insulin, ligands structurally similar to insulin are able to activate insulin receptors, such as insulin growth factors. The same happens when insulin activates the downstream effectors by binding to insulin growth factors. Upon ligand–receptor interaction, signal transduction occurs through kinase domains of IR, IGF1/IR, and IGFR receptors. The main phosphorylation targets are the insulin response elements (IRSs), which further transduce the signal to different signaling cascades, exerting the various effects of insulin and IGFs [34]. These could be metabolic or mitogenic and include the PI3K/SREBP pathway (lipid synthesis), the PI3K/AKT/GSK3/eIF2B pathway (glycogen and protein synthesis and apoptosis regulation), and the RAS/MAPK/ERKS/RSK/ELK1 pathway (gene transcription, protein regulation, cell proliferation, and synaptic plasticity) [34]. Overall, both insulin and *NGF* are able to transduce signals that culminate with the activation of the MEK/ERK pathway. As described above, in DM1, there is a deregulation of this pathway (increase in *ERK1/ERK2* and decrease upon insulin stimulation).

#### 4.1.4. Myogenesis- and Signaling-Associated Pathways 

The majority of genes included in cluster 2 (Figure 6), in particular *MYOG*, *ID1*, *ID2*, and *MYOD1,* play a significant role in skeletal muscle, as mentioned in Table 3. During embryogenesis, mononucleated myoblasts differentiate into myocytes, which fuse to originate multinucleated myotubes to be maturated into multinucleated myofibers that are grouped into highly oriented bundles, forming a single long cylinder (Figure 8). The reorganization of diverse upstream regulators of muscle progenitor differentiation and commitment to the myogenic fate requires the expression of the early myogenic regulatory factors MYF5, MRF4, and *MYOD*, as well as the late differentiation marker *MYOG* [136].

During development, the balance between proliferation and differentiation of myogenic progenitors is tightly regulated to allow for muscle growth while maintaining a pool of undifferentiated progenitors. Several signaling pathways play a role in the switch between proliferation and differentiation of skeletal muscle progenitor cells during development [137]. The major regulator of the muscle progenitor pool is the NOTCH signaling pathway, inhibiting myoblast differentiation in various organisms. Consistent with this idea, NOTCH signaling seems to be prevalent in muscle progenitors and declines throughout differentiation [138]. The NOTCH downstream target split-1 (HES1) is expressed in an oscillatory manner in developing and adult muscle stem cells [139]. Cyclic HES1 expression regulates the downstream target, *MYOD*, the levels of which also oscillate, allowing for the maintenance of the undifferentiated and proliferative state of the stem cells [139]. Additional signaling pathways are also involved in maintaining muscle progenitors in a proliferative state, such as BMP signaling [140]. Inhibitors of DNA-binding (ID) proteins are negative regulators of basic helix–loop–helix transcription factors and generally stimulate cell proliferation and inhibit differentiation. In addition, the BMP target gene inhibitor of DNA-binding protein 1 (*ID1*) is required to maintain satellite cells (SCs) in a proliferative and undifferentiated state [141,142]. In fact, both *ID1* and *ID2* are considered genes that inhibit differentiation through promotion of cell proliferation.

As mentioned above, the regulation of skeletal muscle formation (myogenesis) involves several different signaling pathways that are tightly regulated to control cell differentiation and proliferation [126,143,144]. In DM1, the activity of pathways that promote differentiation is decreased, whereas the activity of pathways that promote proliferation is increased, which significantly impairs and delays myogenesis [69]. Based on our results, one can hypothesize that *ID1* and *ID2* are deregulated in DM1, favoring cell proliferation instead of differentiation and leading to delayed myogenesis, which is consistent with the muscle immaturity observed in patients with DM1. Conversely, *MYOG* expression could be somehow deregulated, impairing the transition of myoblasts to myocytes and decreasing the initiation of the differentiation process. Therefore, we propose four genes as novel target candidates highly relevant to the pathophysiology of DM1 that should be explored in future studies, namely *MYOD, MYOG, ID1*, and *ID2*.

Based on the results of this study, it is clear that the genes identified in the cluster analysis are related to either metabolism and/or skeletal muscle. With regard to DM1, we suggest that genes in clusters 1, 2, and 3 are crucial for the pathophysiological mechanisms of this neuromuscular disease. Analyses have provided insights into the identification of novel metabolic process targets for DM1, as well as potential candidate genes and pathways that might be altered in this disorder and that should be further studied.

## 5. Conclusions

To the best of our knowledge, this is the first study using an automatic text-mining analysis to explore novel metabolism-related targets for DM1. The number of articles in this area has drastically increased in recent years, meaning considerable efforts are being applied to discover the pathophysiological mechanisms underlying DM1. The identified gene candidates could connect aspects of the disease that were previously considered unrelated and could be important to further explore new therapeutic approaches for DM1. However, it needs to be noted that the results reported herein were obtained exclusively using a bioinformatic approach. Therefore, although the novel identified metabolic process targets are a valuable point of reference for potential studies, they need to be further validated on molecular and cellular levels. Overall, the most relevant putative targets identified in the present study are NGF, NTRK1, ROCK1, ROCK2, DAG, ACTA1, ID1, ID2, MyoD, and MyoG. We strongly believe that some of these targets could be druggable for DM1. However, the next step will be the evaluation of these relevant putative targets in DM1 samples (cells, as well as muscle and biofluids) to confirm whether their expression and/or activity is dysregulated in DM1. Subsequently, the modulation of their levels is mandatory to understand whether the disease phenotype can be improved. The genes that improve disease phenotype can be used as novel therapeutic targets for DM1. This work is expected to promote further research and development with respect to the study of the pathophysiological mechanisms of DM1 and to provide guidance for the treatment of DM1 in patients.

## Figures and Tables

**Figure 1 ijerph-20-02283-f001:**
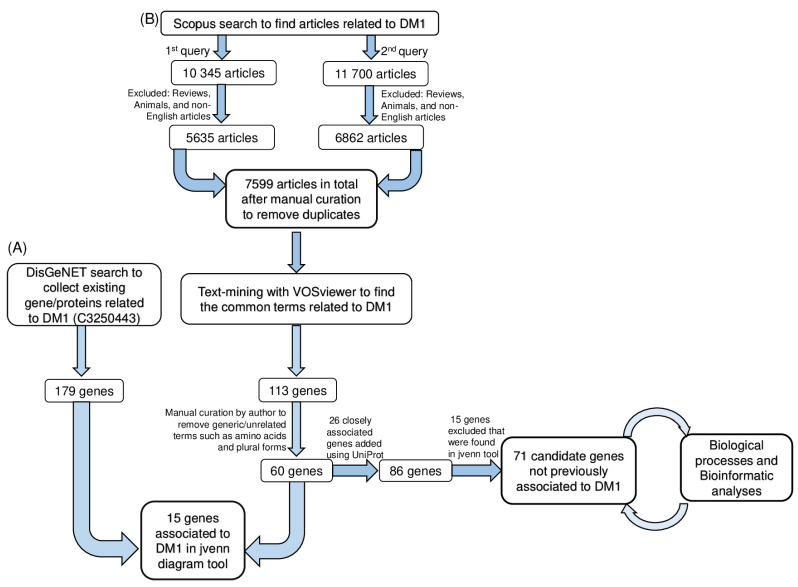
Workflow of the strategy used for literature search and bibliometric analysis to unravel a functional association between myotonic dystrophy type 1 and metabolism. (**A**) DisGeNET-based collection of previously reported molecular associations with DM1. Terminologies corresponding to genes only were selected and compared with DisGeNET data. This analysis yielded a set of 15 genes previously associated with DM1 in DisGeNET. (**B**) Workflow used to obtain the common terms related to metabolism/metabolic and DM1 using VOSviewer software. Following the literature search, text-mining with VOSviewer software was used to uncover the most appropriate terminologies from scientific articles. We also focused on the 71 genes identified in VOSviewer analysis but that were not previously associated with DM1 in DisGeNET. These genes emerged as potential DM1 targets. DM1, myotonic dystrophy type 1.

**Figure 2 ijerph-20-02283-f002:**
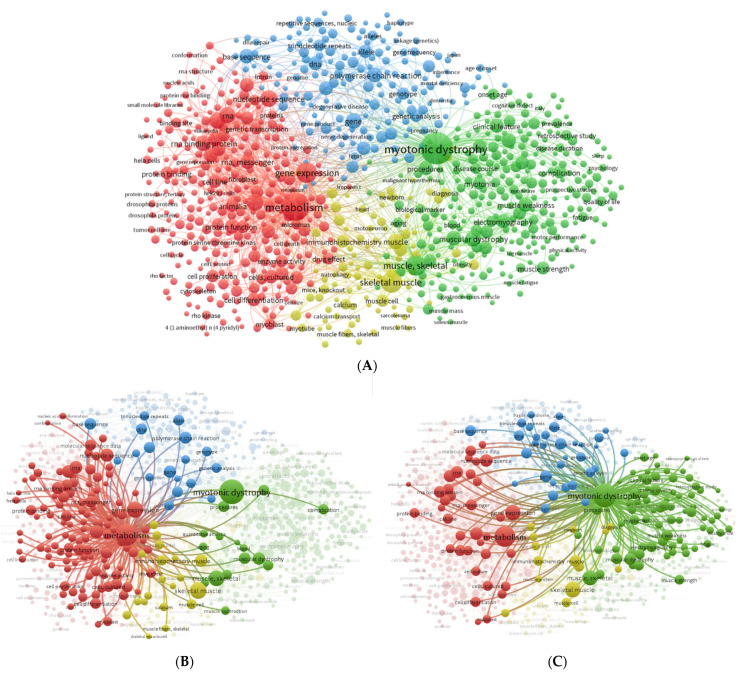
Keyword network visualization map. (**A**) Global network of the co-occurrence of all keywords co-. (**B**) Metabolism co-occurrence network. (**C**) Myotonic dystrophy co-occurrence network. The co-occurrence network visualization map was created using VOSviewer (https://www.vosviewer.com/ (accessed on 21 December 2022). The node size reflects the occurrence of the term, and the edges point out co-occurring terms. The lines demonstrate keywords occurring together. The total link strength shows the number of total publications in which two keywords appear. The size of the nodes reflects the co-occurrence of the keywords, and the distance between two nodes represents the association between the keywords. Terms are grouped into clusters according to research interests, which are represented by different colors: cluster 1, red; cluster 2, green; cluster 3, blue; and cluster 4, yellow.

**Figure 3 ijerph-20-02283-f003:**
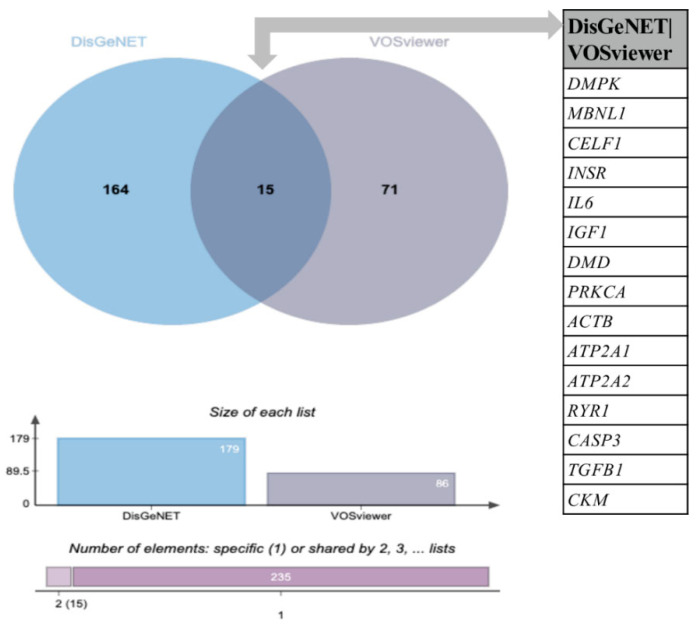
Venn diagram analysis of the DisGeNET and VOSviewer lists of genes. The Venn diagram shows an intersection between the DisGeNET and VOSviewer lists of genes. Each circle corresponds to an entry/gene list. Fifteen (15) common genes were found between the DisGeNET and VOSviewer lists; 164 are exclusive to the DisGeNET list, and 71 are exclusive to the VOSviewer list. The chart presented below the Venn diagram shows the list size and intersection size repartition. A comparative analysis was performed with jvenn software (http://jvenn.toulouse.inra.fr/app/example.html (accessed on 28 November 2022)).

**Figure 4 ijerph-20-02283-f004:**
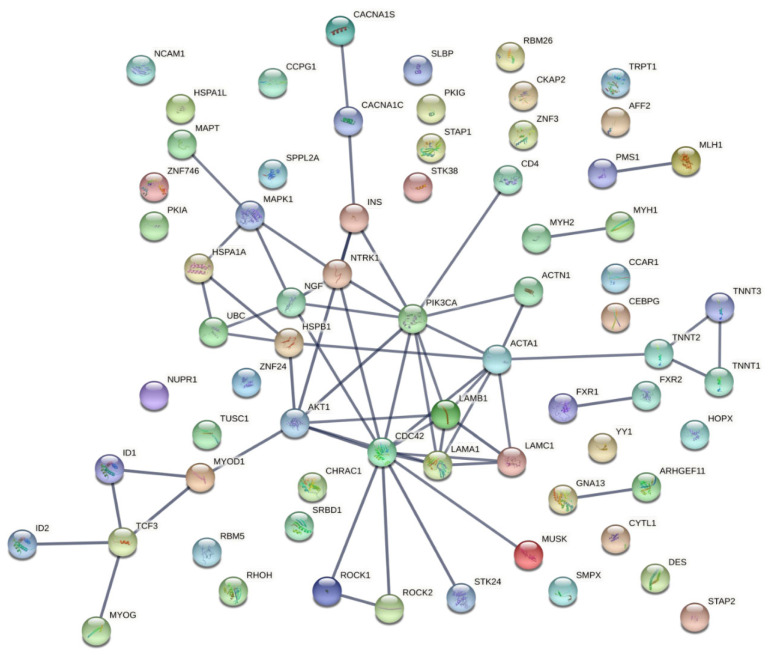
Protein–protein interaction network of the 71 candidate targets for myotonic dystrophy type 1 generated using the STRING database. Network analysis of the 71 genes considering all the possible interactions. The thickness of the lines indicates the degree of confidence of the interaction (wider lines indicate stronger evidence of interactions).

**Figure 5 ijerph-20-02283-f005:**
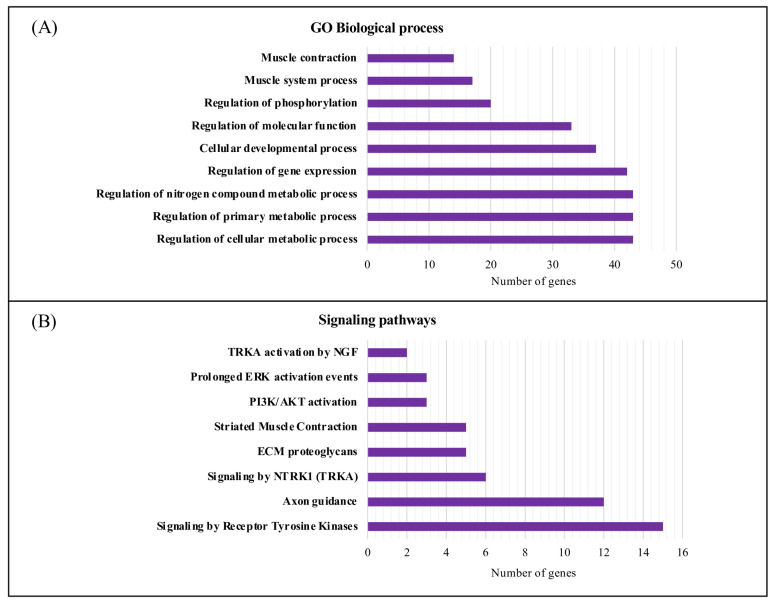
Biological process and signaling pathway functional enrichment analysis of the candidate molecular targets for DM1. (**A**) The top 10 biological processes associated with the candidate molecular targets for DM1. (**B**) REAC signaling pathway enrichment analysis of the eight overrepresented signaling pathways using g:Profiler (https://biit.cs.ut.ee/gprofiler/gost (accessed on 29 August 2022). The purple bars represent the number of genes annotated in the category. GO, gene ontology; NGF, nerve growth factor; ERK, extracellular signal-regulated kinase; PI3K/AKT, phosphatidylinositol-3-kinase/ protein kinase B; ECM, extracellular matrix; *NTRK1* (TRKA), neurotrophic tyrosine kinase receptor type 1.

**Figure 6 ijerph-20-02283-f006:**
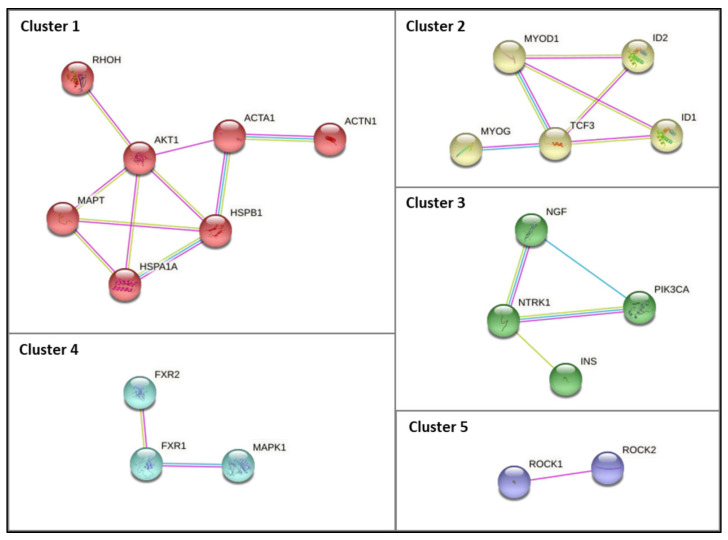
PPI subnetworks of clusters (1 to 5) comprising the genes identified in VOSviewer analysis associated with GO biological processes related to protein metabolism. The PPI subnetwork was constructed using the STRING database by applying the Markov clustering (MCL) algorithm (inflation parameter of 1.8). The edges indicate that the directly linked proteins are part of the same physical complex, although in large complexes, this may not signify that they are directly bound to each other. The edge color denotes the interaction source; blue edges denote known interactions from curated databases, pink edges denote experimentally determined known interactions, and green edges denote interactions inferred by text mining.

**Figure 7 ijerph-20-02283-f007:**
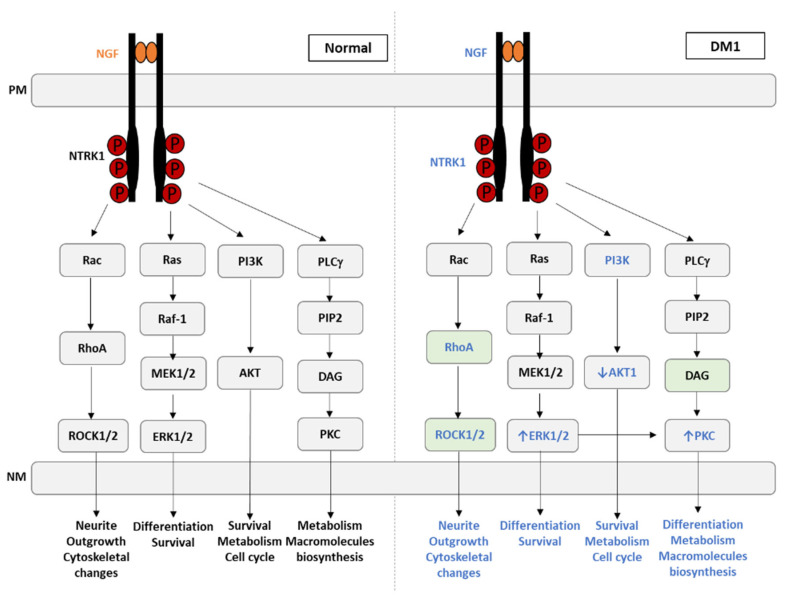
NGF-mediated TRK receptor dimerization and activation of three main intracellular signaling pathways under normal conditions and its deregulation in myotonic dystrophy type 1. The eight genes found in our analysis associated with the TRK signaling pathways are identified in blue, as well as the biological processes affected in DM1. The green boxes indicate the genes never associated with DM1. PM, plasma membrane; NM, nuclear membrane; *NGF*, nerve growth factor; *NTRK1*, neurotrophic receptor tyrosine kinase 1; RAC, Ras-related C3 botulinum toxin substrate 1; RAS, rat sarcoma; *PI3K*, phosphoinositide 3-kinase; PLC-γ, phospholipase C gamma; *RhoA*, Ras homolog family member A; Raf1, Raf-1 proto-oncogene, serine/threonine kinase; *PIP2*, phosphatidylinositol 4,5-bisphosphate; MEK1/2, MAP kinase kinases MEK-1 and MEK-2; AKT, protein kinase B; DAG, diacylglycerol; *ROCK1/2*, Rho-associated protein kinase 1/2; ERK1/2, extracellular signal-regulated protein kinase; PKC, protein kinase C.

**Figure 8 ijerph-20-02283-f008:**
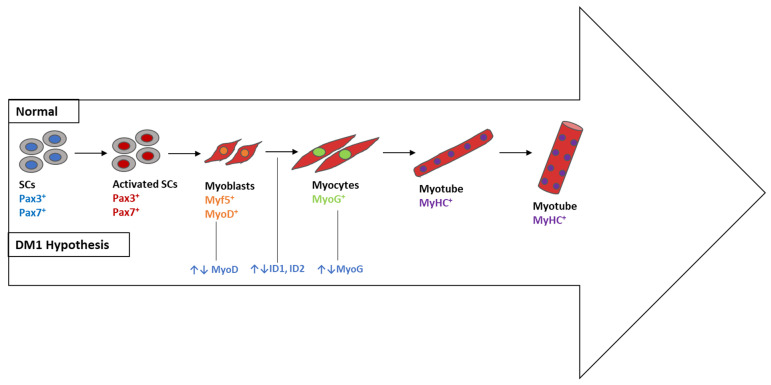
Skeletal muscle differentiation in vitro and its deregulation in DM1 (DM1 hypothesis). The differentiation process starts when Pax3^+^ and/or Pax7^+^ progenitors begin to express Myf5 or *MyoD* as committed myoblasts. The latter gradually express myogenin (*MyoG*) and form single-nucleated nascent myotubes with myosin heavy chain (MHC^+^). The last step consists of myotube fusion to form multinucleated myotubes. SCs, satellite cells; MYF5, myogenic factor 5; MyHC, myosin heavy chain; Pax3/Pax7, paired box transcription factor 3/paired box transcription factor 7; *MyoG*, myogenin; *MyoD*, myogenic differentiation 1; *ID1*, inhibitor of DNA binding 1; *ID2*, inhibitor of DNA binding 2.

**Table 1 ijerph-20-02283-t001:** Top six keywords from the global network visualization map. In the network visualization map, a link expresses a co-occurrence association between keywords. These links have a strength identified by a positive number; a higher number indicates a stronger link. The total link strength indicates the total number of publications in which two keywords appear.

Rank	Keyword	Total Link Strength
1	Myotonic dystrophy	29,037
2	Metabolism	28,505
3	Gene expression	15,044
4	Skeletal muscle	14,660
5	Muscle, skeletal	13,750
6	RNA-binding proteins	10,329

**Table 2 ijerph-20-02283-t002:** Detailed list of common genes identified in DisGeNET and VOSviewer analysis. Proteomics DB (https://www.proteomicsdb.org/(accessed on 21 December 2022) and UniProt (https://www.uniprot.org/ (accessed on 21 December 2022) were used to investigate the biological processes in which these genes participate. More information on their subcellular localization and molecular function are presented in Appendix A.

UniProt Accession Number	Gene Name	Protein Name	Biological Process
Q09013	*DMPK*	Myotonic dystrophy protein kinase	Muscle cell apoptotic processing; intercellular signal transduction; regulation of excitatory postsynaptic membrane potential in skeletal muscle contraction; cellular calcium ion homeostasis
Q9NR56	*MBNL1*	Muscleblind Like Splicing Regulator 1	Neurogenesis; mRNA splicing; mRNA processing; myoblast differentiation
Q92879	*CELF1*	CUGBP Elav-Like Family Member 1	mRNA processing; germ cell development; positive regulation of gene expression and cell death
P06213	*INSR*	Insulin Receptor	Carbohydrate metabolism; activation and positive regulation of protein kinase activity; glucose homeostasis; insulin receptor signaling pathway
P05231	*IL-6*	Interleukin 6	Acute-phase inflammatory response; cellular response to lipopolysaccharide; glucose homeostasis; negative regulation of lipid storage; positive regulation of MAPK cascade
P05019	*IGF-1*	Insulin-like growth factor 1	Myogenesis; ERK1 and ERK2 cascade; glycolate metabolic process; muscle hypertrophy; myoblast differentiation and proliferation
P11532	*DMD*	Dystrophin	Myogenesis; actin cytoskeleton organization; muscle cell development; regulation of muscle system process; skeletal muscle tissue development
P17252	*PRKCA*	Protein Kinase C alpha	Cell adhesion; neurotransmitter; angiogenesis; protein synthesis inhibitor; positive regulation of bone resorption; protein phosphorylation; regulation of mRNA stability
P60709	*ACTB*	Beta-actin	Blood coagulation; rotamase; cell motility; axongenesis
O14983	*ATP2A1*	ATPase Sarcoplasmic/Endoplasmic Reticulum Ca2+ Transporting 1	Blood coagulation; calcium transport; ATP synthesis; maintenance of mitochondrion location; negative regulation of striated muscle contraction
P16615	*ATP2A2*	ATPase Sarcoplasmic/Endoplasmic Reticulum Ca2+ Transporting 2	Cell adhesion; blood coagulation; translocation; transport; ATP synthesis
P21817	*RYR-1*	Ryanodine receptor 1	Calcium transport; muscle contraction; skeletal muscle fiber development
P42574	*CASP3*	Caspase 3	Protease; apoptosis; DNA damage; proteolysis; protein processing
P01137	*TGFB1*	Transforming Growth Factor Beta 1	ATP synthesis; growth arrest; inflammatory response; DNA replication inhibition; neurodegeneration; MAPK cascade
P06732	*CKM*	Creatine kinase, M-type	Creatine metabolic process; phosphocreatine biosynthetic process

**Table 3 ijerph-20-02283-t003:** Summary of the 71 potential novel molecular targets for DM1 that were not previously associated with the disease in DisGeNET and the correspondent VOSviewer occurrence score. ProteomicsDB and UniProt were used to investigate biological processes in which these 71 genes participate.

UniProt ID	Gene/miRNA	Protein	Biological Process	Generic Gene Role/Function in DM1	VOSviewer Occurrences	Reference
Q5T8P6	*RBM26*	RNA-binding protein 26	mRNA processing	- Protein-coding gene critical for PAXT-mediated nuclear RNA.	478	[32]
Q75NE6	*MIR17HG*	Putative microRNA 17 host gene protein	-	- Exhibits complex links to cancer metastasis.	253	[33]
P01398	*INS*	Insulin	Glucose metabolism, sigma factor, transcription	*- INS* is linked with the regulation of muscle protein synthesis, through which reduced insulin sensitivity effects occur in lower muscle mass. - Insulin signaling is known to be a significant contributor to DM1.	196	[34]
P13591	*NCAM1*	Neural cell adhesion molecule 1	Cell adhesion plasma membrane, virus receptor activity	- Its site of expression/function in the membrane protein - Known to occur more frequently in nuclear clump fibers in DM2 than in DM1.	188	[35]
Q9Y6E0	*STK24*	Serine/threonine-protein kinase 24	Protein phosphorylation, signal transduction	- Promotes apoptosis in response to caspase activation and stress stimuli.	162	[36]
Q9UHP9	*SMPX*	Small muscular protein	Striated muscle contraction	- Functions to promote myocyte fusion by increasing the activity of the nuclear factor of activated T cells and MEF2 transcription factors through IGF1 signaling. - Important for muscle fiber organization and distal myopathy.	145	[37]
P68133	*ACTA1*	Actin, alpha skeletal muscle	Muscle contraction, skeletal muscle fiber adaption and development, skeletal muscle thin filament assembly	- Plays a role in the integrity, motility, and structure of all eukaryotic cells. It is also expressed in skeletal muscle. - Mutations in this gene are likely to cause various kinds of myopathy. - *ACTA1* mutations are a notable cause of serious congenital myopathies with no treatment.	128	[38]
Q8TCT8	*SPPL2A*	Signal peptide peptidase-like 2A	Membrane protein ectodomain proteolysis, membrane protein proteolysis	- Plays a key role in the development and purpose of antigen-presenting cells such as dendritic cells. - Shows a druggable pharmacological target, with the potential to provide a novel approach for treating autoimmune diseases by targeting dendritic cells and B cells.	112	[39]
Q8N5C6	*SRBD1*	S1 RNA-binding domain-containing protein 1	Translation, nucleobase-containing compound metabolic process	*- SRBD1* is an RNA-binding protein usually identified in *E.coli*. - Participates maintainence of homeostasis, cell growth, induction of apoptosis, and protein synthesis. - *SRBD1* is known to be sensitive to early-onset normal tension glaucoma, although the functions of this gene in other fields are unclear.	112	[40]
O60356	*NUPR1*	Nuclear protein 1	Detoxification, protein acetylation, skeletal muscle cell differentiation	- Little information about *NUPR1* is available in relation to DM1, nuclear envelope alterations are observed in DM1 primary myoblasts.	101	[41]
P40692	*MLH1*	DNA mismatch repair protein MLH1	Spermatogenesis, oogenesis, mismatch repair	- Provides instructions to make proteins that play a role in repairing DNA.	92	[42]
Q13464	*ROCK1*	Rho-associated protein kinase 1	Regulation of cell adhesion, signal transduction	- Destabilizes the actin cytoskeleton via regulation of myosin light chain 2 (MLC2) phosphorylation.	89	[43]
P41134	*ID1*	DNA-binding protein inhibitor ID-1	DNA-directed RNA polymerase transcription, apoptosis, angiogenesis, antibiotic resistance	- Helix–loop–helix protein involved in cellular growth and processes that has no basic region and does not bind to DNA. - *ID1* inhibits skeletal muscle but has no direct association with DM1.	78	[44]
P10636	*MAPT*	Microtubule-associated protein tau	rRNA metabolic process, protein homo-oligomerization	- Contributes to microtubule assembly and stability.	78	[45]
P12882	*MYH1*	Myosin-1	Muscle contraction	- Myosin changes chemical energy into mechanical energy via the hydrolysis of ATP.	77	[46]
P51114	*FXR1*	Fragile X mental retardation syndrome-related protein 1	Apoptotic process, muscle organ development, skeletal muscle organ development	*- FXR1* splicing is key for muscle development and biomolecular condensates in muscle cells.	74	[47]
P01138	*NGF*	Beta nerve growth factor	Apoptosis, inflammatory response	- Provides directions for the manufacture of a protein called nerve growth factor beta (NGFβ). - Contributes to the development and survival of nerve cells.	71	[48]
Q86TN4	*TRPT1*	tRNA 2’-phosphotransferase 1	tRNA processing, regulation of protein kinase activity	- Predicted to enable tRNA 2’-phosphotransferase activity. - Anticipated to be involved in tRNA splicing through ligation and endonucleolytic cleavage.	68	[49]
P15923	*TCF3*	Transcription factor E2-alpha	DNA-directed RNA polymerase, transcription, transcription, regulation, apoptosis	- Plays a vital role in embryogenesis.	66	[50]
P31749	*AKT1*	Protein kinase B	Cell differentiation, glucose metabolic process, protein kinase B signaling, protein phosphorylation	*- AKT1* regulates processes including cell survival, metabolism, and angiogenesis. This gene plays a role in various signaling pathways in neurodegenerative diseases. - Impairment of AKT signaling in DM1 pathogenesis causes surges in apoptosis and autophagy, which can be affected in DM1 by AMPK downregulation.	65	[51]
Q9BPY8	*HOPX*	Homeodomain-only protein	DNA-directed RNA polymerase, Transcription, regulation of protein binding	- Involved in the regulation of growth and differentiation of myocytes, stem cells, and lymphocytes.	61	[52]
Q14344	*GNA13*	Guanine nucleotide-binding protein subunit alpha-13	Transduction, blood coagulation, differentiation, cell shape	- Contributes to bone homeostasis, angiogenesis, platelet activation, and localization of germinal B cells.	60	[53]
P17028	*ZNF24*	Zinc finger protein 24	DNA-directed RNA polymerase, transcription, myelination	- Controls proliferation, differentiation, and migration in many kinds of cells.	60	[54]
P01730	*CD4*	T-cell surface glycoprotein CD4	Transduction, cell adhesion, innate immunity	*- CD4* (T cells) fights infection. - Crucial role in adaptive immune responses, such as the stimulation of cytotoxic lymphocytes. - HIV particles are reproduced by infected CD4 T cells.	57	[55]
P28482	*MAPK1*	Mitogen-activated protein kinase 1	DNA-directed RNA polymerase, cytosine metabolic process, ERK1 and ERK2 cascade, protein phosphorylation	- Involved in muscular dystrophies. - Many researchers have reported the activation of extracellular signal-related kinases (ERKs), c-Jun N-terminal kinases (JNKs), and p-38 MAPK in skeletal and cardiac muscle of Duchenne muscular dystrophies. - ERKs were also found to be deregulated in DM1. - MAPKs are targets of PKC proteins.	55	[56,57]
Q9ULG6	*CCPG1*	Cell cycle progression protein 1	Positive regulation of cell cycle, positive regulation of cell population proliferation	- Cell cycle progression gene 1 and has no known physiological role. - This gene is important for ER-phagy and pancreatic ER proteostasis.	53	[58]
P17661	*DES*	Desmin	Cytoskeleton organization, muscle contraction	- A class III intermediate filament used as a marker of myogenic cells. - Found in cardiac and skeletal muscle.	53	[59]
Q15669	*RHOH*	Rho-related GTP-binding protein RhoH	Sigma factor, transcription regulation	- Rho genes are part of the Ras superfamily of GTPases. - Alteration of signal transduction by Rho GTPases is a repeated theme in the flow of human malignancies.	48	[60]
P60953	*CDC42*	Cell division control protein 42 homolog	Actin cytoskeleton organization, actin filament branching	- Plays an essential role in local F-actin organization by several kinase and non-kinase effector proteins - Cell cycle regulator.	45	[61]
Q9NRG0	*CHRAC1*	Chromatin accessibility complex protein 1	Nucleosome assembly and mobilization, regulation of DNA replication	- A potential target for lung cancer.	45	[62]
P15172	*MYOD1*	Myoblast determination protein 1	Muscle organ development, positive regulation of muscle cell differentiation, protein phosphorylation	- Encodes a nuclear protein and is known to be involved in muscle regeneration and differentiation.	45	[63]
Q2TAM9	*TUSC1*	Tumor suppressor candidate gene 1 protein	-	- Known to exhibit tumor-suppressor activity as a candidate tumor suppressor gene.	44	[64]
Q13698	*CACNA1S*	Voltage-dependent L-type calcium channel subunit alpha-1S.	Muscle contraction, calcium ion import and transport	- Delivers directions to make a subunit of a structure called a calcium channel.	43	[65]
P15173	*MYOG*	Myogenin	Positive regulation of muscle atrophy, regulation of myoblast fusion, skeletal muscle cell differentiation	- Plays a role in muscle atrophy, muscle differentiation, and embryonic skeletal fiber muscle differentiation.	43	[66]
P0CG48	*UBC*	Polyubiquitin-C	Protein ubiquitination	- The ubiquitin (Ub) system plays a vital role in protein homeostasis.	43	[67]
P0DMV8	*HSPA1A*	Heat shock 70 kDa protein 1A	ATP metabolic process, cellular response to heat, lysosomal transport, protein refolding and stabilization	- Important role in the protein quality control system. - Ensures the correct folding of proteins and controls the targeting of proteins for upcoming degradation.	42	[68]
P45378	*TNNT3*	Troponin T, fast skeletal muscle	Muscle contraction, skeletal muscle contraction	- Provides instructions to make a protein called troponin T. - Troponin T is found in skeletal muscles, which are used for movement. - This gene is linked to impaired muscle function.	42	[69]
Q9NRR1	*CYTL1*	Cytokine-like protein 1	Signaling receptor binding, extracellular space, cartilage homeostasis	- Features of a secretory protein. - Linked with conditions such as smoking, cardiac fibrosis, and various tumors.	40	[70]
P61925	*PKIA*	cAMP-dependent protein kinase inhibitor alpha	Negative regulation of protein import into the nucleus	- Protein kinase inhibitors are widely referenced in the literature. - However, because the *DMPK* gene encodes for myotonic dystrophy protein kinase, a serine–threonine kinase with similarity to adenosine monophosphate-dependent protein kinases undergoes self-phosphorylation, in contrast to other membrane proteins.	40	[71]
P12814	*ACTN1*	Alpha-actinin-1	Actin cytoskeleton organization, focal adhesion assembly, muscle cell development	- Cytoskeletal proteins known to exhibit non-muscle functions. - Important for glioma cell motility.	39	[72]
O15085	*ARHGEF11*	Rho guanine nucleotide exchange factor 11	Striated muscle contraction, G-protein-coupled receptor	- Promotes tumor metastasis in glioblastoma and ovarian cancer.	38	[73]
P42336	*PIK3CA*	Phosphatidylinositol 4,5-bisphosphate 3-kinase catalytic subunit alpha isoform	Glucose metabolism, response to muscle stretching, positive regulation of smooth muscle cell proliferation	- Produces an enzyme called PI3K, which has been associated with DM1 - Participates in cellular signaling in response to different growth factors. - The *PIK3CA* gene mainly plays a role in breast cancer and causes mutations in different human malignancies.	38	[57,74]
P53567	*CEBPG*	CCAAT/enhancer-binding protein gamma	mRNA metabolic process, positive regulation of DNA binding and repair	- Potential biomarkers for cancer prognosis. - Plays a role in gastric cancer progression and breast cancer cell migration.	36	[75]
P11047	*LAMC1*	Laminin subunit gamma 1	Cell adhesion, tissue development	- Associated with the development and occurrence of tubal cancer and other malignant tumors. - Its molecular mechanism remains unclear.	36	[76]
P51816	*AFF2*	AF4/FMR2 family member 2	mRNA processing, RNA splicing, regulation of gene expression	- Encodes a recognized transcriptional activator. - conserved and expressed in the human brain.	ND	[77]
Q13936	*CACNA1C*	Voltage-dependent L-type calcium channel subunit alpha-1C	Cardiac muscle cell action potential involved in contraction, cardiac conduction	- Encodes calcium channels in heart tissue and its gain-of-purpose mutations in arrhythmias and sudden death. - *CACNA1C* and GJA1 upregulation may contribute to cardiac impairment observed in DM1 patients.	ND	[78]
Q8IX12	*CCAR1*	Cell division cycle and apoptosis regulator protein 1	DNA-directed RNA polymerase, transcription, sigma factor, transcription, Regulation, mRNA splicing	- Functions as a key proliferation-inducing gene and p35 coactivator.	ND	[79]
Q8WWK9	*CKAP2*	Cytoskeleton-associated protein 2	Apoptotic process, mitotic cytokinesis	- A powerful microtubule growth factor. - Plays important roles as an oncogene and spindle protein and in proliferation.	ND	[80]
P51116	*FXR2*	Fragile X mental retardation syndrome-related protein 2	Regulation of mRNA stability, positive regulation of protein phosphorylation	*- FXR2* is an RNA-binding protein known to play a role in fragile X cognitive disability syndrome.	ND	[81]
P34931	*HSPA1L*	Heat shock 70 kDa protein 1-like	Unfolded protein response, vesicle-mediated transport	- Encodes a 70kDa heat shock protein. - Ensures protein quality control of the cell.	ND	[82]
P04792	*HSPB1*	Heat shock protein beta 1	Regulation of protein phosphorylation, chaperon-mediated protein folding	- Preserves cellular proteostasis during the course of stress conditions.	ND	[83]
Q02363	*ID2*	DNA-binding protein inhibitor ID-2	Developmental protein, regulation of lipid metabolic process, cellular senescence	- Regulator of developmentally associated genes and tumor growth in vitro, as well as in vivo, in Ewing sarcoma tumors. - Involved in cellular growth.	ND	[84]
P25391	*LAMA1*	Laminin subunit alpha 1	Cell adhesion, tissue development, protein phosphorylation	- Encodes the alpha 1 subunits of laminin. - Implicated in various biological processes such as migration, cell adhesion, signaling, differentiation, and metastasis. - Mutations in this gene may be related to Poretti-Boltshauser syndrome.	ND	[85]
P07942	*LAMB1*	Laminin subunit beta 1	Cell adhesion, cell migration, tissue development	- Plays significant roles in different kinds of tumors, including breast cancer, glioblastoma multiforme, prostate cancer, and hepatocellular carcinoma.	ND	[86]
P43246	*MLH2*	DNA mismatch repair protein Msh2	DNA repair, oxidative phosphorylation, mismatch repair	- Delivers instructions for making a protein that play an important role in DNA repair.	ND	[87]
O15146	*MUSK*	Muscle, skeletal receptor tyrosine-protein kinase	Skeletal muscle acetylcholine-gated channel clustering	- Encodes a muscle-specific tyrosine kinase receptor that plays a role in clustering of the acetylcholine receptor in postsynaptic neuromuscular connections. - Mutations in this gene have been related to congenital myasthenic syndrome.	ND	[88]
Q9UKX2	*MYH2*	Myosin-2	Muscle contraction, muscle filament sliding	- Needed for cytoskeleton organization and muscle contraction.	ND	[89]
P04629	*NTRK1*	High affinity nerve growth factor receptor	Aging, nerve growth factor signaling pathway	*NTRK* plays a role in the growth and normal functioning of the nervous system. However, tropomyosin receptor kinase (trk) is involved in different kinds of cancer.	ND	[90]
Q9Y2B9	*PKIG*	cAMP-dependent protein kinase inhibitor gamma	Negative regulation of protein import into the nucleus	- Protein kinase inhibitors are broadly referenced in the literature. - However, because the *DMPK* gene encodes for myotonic dystrophy protein kinase, a serine–threonine kinase with similarity to adenosine monophosphate-dependent protein kinases undergoes self-phosphorylation, in contrast to other membrane proteins.	ND	[71]
P52756	*RBM5*	RNA-binding protein 5	Apoptosis, mRNA splicing	- Acts as a tumor suppressor. - Controls cell growth, cell cycle progression, and apoptosis in cell homeostasis. - *RBM5* expression may be a potential curative target for drug-resistant lung cancer.	ND	[91]
O75116	*ROCK2*	Rho-associated protein kinase 2	Actin cytoskeleton organization, positive regulation of MAPK cascade	- Stabilizes actin cytoskeleton via regulation of cofilin phosphorylation.	ND	[43]
Q14493	*SLBP*	Histone RNA hairpin-binding protein	mRNA processing, mRNA transport	- Essential for the coordinate synthesis of DNA and histone proteins. - Needed for progression through the cell division cycle.	ND	[92]
Q9ULZ2	*STAP1*	Signal-transducing adaptor protein 1	Transduction, positive regulation of gene expression	- Candidate gene related to familial hypercholesterolemia. - *STAP1* is observed in immune cells.	ND	[93]
Q9UGK3	*STAP2*	Signal-transducing adaptor protein 2	---	- Regulates different intercellular signaling pathways and promotes prostate cancer flow via EGFR activation.	ND	[94]
Q15208	*STK38*	Serine/threonine-protein kinase 38	Protein phosphorylation	- Negative regulator of MAPK1/2 signaling.	ND	[95]
P04637	*TB53*	Cellular tumor antigen p53	Cell aging, protein stabilization, regulation of cell cycle	- Acts as a tumor suppressor in various tumor types. - Regulates in DNA repair, autophagy, senescence, and cell cycle arrest.	ND	[96]
P13805	*TNNT1*	Troponin T, slow skeletal muscle	Muscle contraction, skeletal muscle contraction	*- TNNT1* is known as the slow skeletal troponin T subunit. - Mutations in this gene lead to nemaline myopathy type 5. - Causes most troponin-related skeletal myopathies.	ND	[97]
P45379	*TNNT2*	Troponin T, cardiac muscle	Muscle contraction, muscle filament sliding, actin crosslink formation	- It is unclear whether *TNNT2* mis-splicing can be considered a specific cardiac biomarker in adult skeletal muscles of DM1 patients, suggesting that alternative splicing of this gene may be useful as a cardiac biomarker.	ND	[98]
P25490	*YY1*	Transcriptional repressor protein YY1	DNA-directed RNA polymerase, transcription, differentiation spermatogenesis, DNA damage	- Involved in regulating the expression of a large number of genes and binds to interferon-beta (IFN-β) promoter to either repress or activate its expression. - It could play a role in modulating the cellular response to dsRNA.	ND	[99]
P17036	*ZNF3*	Zinc finger protein 3 (isoform)	Cell differentiation, leukocyte activation	- Serves as a specific RNA-binding domain, which helps *MBNL* to identify various target mRNAs by binding to a wide area of motifs.	ND	[100]
Q6NUN9	*ZNF746*	Zinc finger protein 746	DNA-directed RNA polymerase, transcription	- Performs the function of promoting the occurrence of hepatocellular carcinoma. - Recognized as a substrate of E3 ligase Parkin, and its accumulation is related to Parkinson’s disease.	ND	[101]

Abbreviations: ND—the score was not available, given these were added by gene proximity.

## Data Availability

The data used to support the findings of this study are included within the article/Appendix A. Further inquiries can be directed to the corresponding author.

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
