# Peer review of "Automatic Text-Mining Approach to Identify Molecular Target Candidates Associated with Metabolic Processes for Myotonic Dystrophy Type 1"

_ijerph, 2023, doi:10.3390/ijerph20032283_

Round 1
Reviewer 1 Report
This paper reported an in-silicon work of searching for molecular target candidates associated with metabolic process for Myotonic Dystrophy type 1. The author identified 71 genes not previously reported yet associated with DM1. They also identified a number of signaling pathways related with DM1. Overall, it is an interesting work and provides insights to the drug design of DM1. Here are my concerns.
1) All the discoveries in this work were based on data analysis and prediction. Please rank the genes and pathways according to the confidence estimation.
2) The authors discovered a batch of genes related with DM1. It is still not clear how to regulate the genes in order to cure DM1. Please provide more specific analysis to those discoveries.
3) If possible , please discuss the drugable genes or proteins for DM1.
Author Response
Reviewer 1
This paper reported an in-silicon work of searching for molecular target candidates associated with metabolic process for Myotonic Dystrophy type 1. The author identified 71 genes not previously reported yet associated with DM1. They also identified a number of signalling pathways related with DM1. Overall, it is an interesting work and provides insights to the drug design of DM1. Here are my concerns.
Authors: Thank you for the positive and encouraging comments about our manuscript.
Reviewer 1: 1) All the discoveries in this work were based on data analysis and prediction. Please rank the genes and pathways according to the confidence estimation.
Authors: We would like to thank the reviewer for the excellent suggestions. We effectively rank our gene list according the VOSviewer occurrences score, and these results were included in Table 3 (new column). Further a new Supplementary Table S8 was included, where the genes were presented according to the ranking. Regarding the signalling pathways ranking these results are already available in the manuscript through the Figure 5B. The P-values (adjusted P-value) are also available in Table S7.
Reviewer 1: 2) The authors discovered a batch of genes related with DM1. It is still not clear how to regulate the genes in order to cure DM1. Please provide more specific analysis to those discoveries.
Authors: Thank you for the valuable comment.
Having in mind the results obtained in the VOSviewer occurrences rank (Table 3 and Supplementary Table S8), the biological processes and signalling pathways enrichment analysis (Figure 5 and Supplementary Table S6 and S7) and finally the results of PPI subnetwork of clusters we realize that the NGF and TRK signalling pathways are among the most relevant signalling pathways obtained in the present study where NGF, RhoA, NTRK1, ROCK1, ROCK2 and DAG are among the top relevant putative targets as highlighted in the discussion section. Additionally, the crosstalk between the NGF and INS mediated signalling is also of particular interest. Regarding the muscle function, including muscle contraction and myogenesis, other top relevant putative targets were identified namely ACTA1, ID1, ID2, MyoD and MyoG.
In the present manuscript we collect evidence of genes and signalling pathways deregulated in DM1. The next step will be the evaluation of the relevant putative targets in DM1 samples (cells, muscle and fluids) to confirm if their expression in DM1 is dysregulated (downregulated, upregulated or normal). The next step will be the modulation of its levels to understand if the disease phenotype can be improved. If so, these genes could be druggable and strategies like the use of siRNAs, and KO mice would be used to downregulate the gene of interest. Additionally, the use of KI mice or overexpression strategies could be used to upregulate the gene of interest.
Reviewer 1: 3) If possible, please discuss the drugable genes or proteins for DM1.
Authors: As discussed in the previous point, the top relevant putative targets are NGF, NTRK1, RhoA, ROCK1, ROCK2, DAG, ACTA1, ID1, ID2, MyoD and MyoG. We strongly believe that some of these could be druggable upon several preliminary tests using DM1 samples. Given the importance of this issue, we decided to include this idea in the conclusion section as follows:
‘In overall, the top relevant putative targets identified in the present study are NGF, NTRK1, ROCK1, ROCK2, DAG, ACTA1, ID1, ID2, MyoD and MyoG. We strongly believe that some could be druggable for DM1. However, the next step will be the evaluation of these relevant putative targets in DM1 samples (cells, muscle and biofluids) to confirm if their expression and/or activity is in fact dysregulated in DM1. Subsequently, the modulation of its levels is mandatory to understand if the disease phenotype can be improved. The genes that improve disease phenotype will be novel therapeutic targets for DM1.’

Reviewer 2 Report
In the manuscript, the authors propose a text-mining approach to analyze literature on myotonic dystrophy type 1. The approach resorts to a bibliometrics analysis and identifies and further analyzes articles regarding such context. Molecular targets are extracted and then compared with genes previously associated with myotonic dystrophy type 1. A thorough discussion is presented on the results.
The manuscript is well written and flows well. The approach combines a quantitative analysis with a more qualitative one. There are few aspects the authors could address to improve the quality of the manuscript. These are given below:
- What is the target of the text mining analysis? Did the authors apply text mining on the abstract of the considered papers?
- The results of query 1 and query 2 it is not very clear. Both queries contain 'metabolic' as first term in the OR clause; therefore, it is fair to imagine that the queries could retrieve similar results. However, in Figure 1 (B) this does not seem the case. Did the authors remove the overlap (if any) between the two queries?
- What does the edge thickness in Figure 2 represent?
- How is link strength defined?
- Discussion is far too long and it is not straightforward to follow as it is. I suggest the authors to organize it in paragraphs or at least in subsections.
- A novel method for cluster analysis could be also used in the context studied in the paper. The authors could consider to add the following work to the references [https://doi.org/10.1016/j.compbiomed.2016.07.015].
Author Response
Reviewer 2
In the manuscript, the authors propose a text-mining approach to analyze literature on myotonic dystrophy type 1. The approach resorts to a bibliometrics analysis and identifies and further analyzes articles regarding such context. Molecular targets are extracted and then compared with genes previously associated with myotonic dystrophy type 1. A thorough discussion is presented on the results.
The manuscript is well written and flows well. The approach combines a quantitative analysis with a more qualitative one.
Authors: Thank you for the very positive comments about our manuscript.
Reviewer 2: There are few aspects the authors could address to improve the quality of the manuscript. These are given below:
- What is the target of the text mining analysis? Did the authors apply text mining on the abstract of the considered papers?
Authors: The main target of text mining analysis was the title, the abstract and keywords selected by the authors of the considered papers.
Reviewer 2: The results of query 1 and query 2 it is not very clear. Both queries contain 'metabolic' as first term in the OR clause; therefore, it is fair to imagine that the queries could retrieve similar results. However, in Figure 1 (B) this does not seem the case. Did the authors remove the overlap (if any) between the two queries?
Authors: We thank the reviewer for this question. In fact, we check all the analysis performed and we realize that by mistake the duplicates were not removed. In this revised version of the manuscript the duplicated papers were removed and the Figure 1, Figure 2 and Table 2 were corrected and new images were included in the manuscript.
Reviewer 2: What does the edge thickness in Figure 2 represent?
Authors: In each keyword network visualization map the edge thickness is identical and there is no meaning for edge thickness (example Figure 2A). When we presented for instance the metabolism co-occurrences network (Figure 2 B) the edge thickness is higher compared to Figure 2A because is a highlight, but again between the network the edge thickness is identical and there is no meaning for edge thickness.
Reviewer 2: How is link strength defined?
Authors: The total link strength shows the number of total publications in which two key words appear.
Reviewer 2: Discussion is far too long and it is not straightforward to follow as it is. I suggest the authors to organize it in paragraphs or at least in subsections.
Authors: Thank you for the great suggestion. The discussion was reorganized and subsections were included.
Reviewer 2: A novel method for cluster analysis could be also used in the context studied in the paper. The authors could consider to add the following work to the references [https://doi.org/10.1016/j.compbiomed.2016.07.015].
Authors: Thank you for bring our attention to this novel method for cluster analysis. Hopefully, we will use it in a near future. Therefore, we decided to include the following sentence and reference:
‘Additionally, a novel automated string-based approach for cluster analysis could also be used as an alternative and interesting approach to the presently used in the manuscript (Cauteruccio et al 2016).’

Round 2
Reviewer 2 Report
The authors correctly addressed my concerns. In my opinion, the manuscript now has achieved a publication-ready quality.